# FigureQA: An Annotated Figure Dataset for Visual Reasoning

## Abstract

We introduce FigureQA, a visual reasoning corpus of over one million question-answer pairs grounded in over $100,000$ images. The images are synthetic, scientific-style figures from five classes: line plots, dot-line plots, vertical and horizontal bar graphs, and pie charts. We formulate our reasoning task by generating questions from 15 templates; questions concern various relationships between plot elements and examine characteristics like the maximum, the minimum, area-under-the-curve, smoothness, and intersection. To resolve, such questions often require reference to multiple plot elements and synthesis of information distributed spatially throughout a figure. To facilitate the training of machine learning systems, the corpus also includes side data that can be used to formulate auxiliary objectives. In particular, we provide the numerical data used to generate each figure as well as bounding-box annotations for all plot elements. We study the proposed visual reasoning task by training several models, including the recently proposed Relation Network as strong baseline. Preliminary results indicate that the task poses a significant machine learning challenge. We envision FigureQA as a first step towards developing models that can intuitively recognize patterns from visual representations of data.

## 1 Introduction

Scientific figures compactly summarize valuable information. They depict patterns like trends, rates, and proportions, and enable humans to understand these concepts intuitively at a glance. Because of these useful properties, scientific papers and other documents often supplement textual information with figures. Machine understanding of this structured visual information could assist human analysts in extracting knowledge from the vast documentation produced by modern science. Besides immediate applications, machine understanding of plots is interesting from an artificial intelligence perspective, as most existing approaches simply revert to inverting the visualization pipeline (i.e. by reconstructing the source data). Mathematics exams, e.g. Graduate Records Examinations (GREs), often include questions about a plot regarding relations between the plot elements. When solving these exam questions, humans don't always build a table of coordinates for all data points, but judge by visual intuition.

Thus motivated, and inspired by recent research in *Visual Question Answering (VQA)* (Antol et al., 2015; Goyal et al., 2016) and *relational reasoning* (Johnson et al., 2016; Suhr et al., 2017), we introduce *FigureQA*. FigureQA is a corpus of over one million question-answer pairs grounded in over $100,000$ figures, devised to study aspects of comprehension and reasoning in machines. There are five common figure types represented in the corpus, which model both continuous and categorical information: *line*, *dot-line*, *vertical* and *horizontal bar*, and *pie plots*. Questions concern *one-to-all* and *one-to-one* relations among plot elements: for example, *Is X the low median?*, *Does X intersect Y?*. Their successful resolution requires inference over multiple plot elements. There are 15 question types in total, which address properties like *magnitude*, *maximum*, *minimum*, *median*, *area-under-the-curve*, *smoothness*, and *intersections*. Each question is posed such that its answer is either *yes* or *no*.

FigureQA is a synthetic corpus, like the related CLEVR dataset for visual reasoning (Johnson et al., 2016). While this means that the data may not exhibit the same richness as figures "in the wild", it permits greater control over the task's complexity, enables auxiliary supervision signals and most

importantly provides reliable ground-truth answers. Further, by analyzing the performance on real figures of models trained on FigureQA it will be possible to extend the corpus to address limitations not considered during generation. The FigureQA corpus can be extended iteratively, each time raising the task complexity, as model performance increases. This is reminiscent of *curriculum learning* (Bengio et al., 2009) allowing iterative pretraining on increasingly challenging versions of the data. By releasing the data now, we want to gauge the interest in the research community and adapt future versions based on feedback, hopefully accelerating research in this direction. Additional annotation is provided to allow researchers to define other tasks than the one we introduce in this manuscript.

The corpus is built using a two-stage generation process. First, we sample numerical data according to a carefully tuned set of constraints and heuristics designed to make sampled figures appear natural. Next we use the *Bokeh* open-source plotting library (Bokeh Development Team, 2014) to plot the data in an image. This process necessarily gives us access to the quantitative data presented in the figure. We also modify the Bokeh backend to output bounding boxes for all plot elements: data points, axes, axis labels and ticks, legend tokens, etc. We provide the underlying numerical data and the set of bounding boxes as supplementary information with each figure, which may be useful in formulating auxiliary tasks like reconstructing quantitative data given only a figure. Bounding box targets of plot elements relevant for answering a question might be useful for supervising an attention mechanism to ignore potential distractors. Experiments in that direction are outside of the scope of this manuscript, but we want to facilitate research of such approaches by releasing these annotations.

As part of the generation process we balance the ratio of yes and no answers for each question type and each figure. This makes it more difficult for models to exploit biases in answer frequencies while ignoring visual content.

We review related work in Section 2. In Section 3 we describe the FigureQA dataset and the visual-reasoning task in detail. Section 4 describes and evaluates four neural baseline models trained on the corpus: a text-only Long Short-Term Memory (LSTM) model (Hochreiter & Schmidhuber, 1997) as a sanity check for biases, the same LSTM model with added Convolutional Neural Network (CNN) image features (LeCun et al., 1998; Fukushima, 1988), and a Relation Network (RN) (Santoro et al., 2017), a strong baseline model for relational reasoning.

The RN respectively achieves accuracies of 72.40% and 76.52% on the FigureQA test set with alternated color scheme (described in Section 3.1) and the test set without swapping colors. An "official" version of the corpus is publicly available as a benchmark for future research.[1] We also provide our generation scripts, which are easily configurable, enabling researchers to tweak generation parameters to produce their own variations of the data.

## 2 RELATED WORK

Machine learning tasks that pose questions about visual scenes have received great interest of late. For example, Antol et al. (2015) proposed the VQA challenge, in which a model seeks to output a correct natural-language answer $a$ to a natural-language question $q$ concerning image $I$. An example is the question "Who is wearing glasses?" about an image of a man and a woman, one of whom is indeed wearing glasses. Such questions typically require capabilities of vision, language, and common-sense knowledge to answer correctly. Several works tackling the VQA challenge observe that models tend to exploit strong linguistic priors rather than learning to understand visual content. To remedy this problem, Goyal et al. (2016) introduced the balanced VQA task. This features triples $(I', q, a')$ to supplement each image-question-answer triple $(I, q, a)$, such that $I'$ is similar to $I$ but the answer given $I'$ and the same $q$ is $a'$ rather than $a$.

Beyond linguistic priors, another potential issue with the VQA challenges stems from their use of real images. Images of the real world entangle visual-linguistic reasoning with common-sense concepts, where the latter may be too numerous to learn from VQA corpora alone. On the other hand, synthetic datasets for visual-linguistic reasoning may not require common sense and may permit the reasoning challenge to be studied in isolation. CLEVR (Johnson et al., 2016) and NLVR (Suhr et al., 2017) are two such corpora. They present scenes of simple geometric objects along with

---

[1]It can be downloaded at project website.

questions concerning their arrangement. To answer such questions, machines should be capable of spatial and relational reasoning. These tasks have instigated rapid improvement in neural models for visual understanding (Santoro et al., 2017; Perez et al., 2017; Hu et al., 2017). FigureQA takes the synthetic approach of CLEVR and NLVR for the same purpose, to contribute to advances in figure-understanding algorithms.

The figure-understanding task has itself been studied previously. For example, Siegel et al. (2016) present a smaller dataset of figures extracted from research papers, along with a pipeline model for analyzing them. As in FigureQA, they focus on answering linguistic questions about the underlying data. Their FigureSeer corpus contains $60,000$ figure images annotated by crowdworkers with the plot-type labels. A smaller set of 600 figures comes with richer annotations of axes, legends, and plot data, similar to the annotations we provide for all $140,000$ figures in our corpus. The disadvantage of FigureSeer as compared with FigureQA is its limited size; the advantage is that its plots come from real data. The questions posed in FigureSeer also entangle reasoning about figure content with several detection and recognition tasks, such as localizing axes and tick labels or matching line styles with legend entries. Among other capabilities, models require good performance in optical character recognition (OCR). Accordingly, the model presented by Siegel et al. (2016) comprises a pipeline of disjoint, off-the-shelf components that are not trained end-to-end.

Poco & Heer (2017) propose the related task of recovering visual encodings from chart images. This entails detection of legends, titles, labels, etc., as well as classification of chart types and text recovery via OCR. Several works focus on data extraction from figures. Tsutsui & Crandall (2017) use convolutional networks to detect boundaries of subfigures and extract these from compound figures; Jung et al. (2017) propose a system for processing chart images, which consists of figure-type classification followed by type-specific interactive tools for data extraction. Also related to our work is the corpus of Cliche et al. (2017). There, the goal is automated extraction of data from synthetically generated scatter plots. This is equivalent to the data-reconstruction auxiliary task available with FigureQA.

FigureQA is designed to focus specifically on reasoning, rather than subtasks that can be solved with high accuracy by existing tools for OCR. It follows the general VQA setup, but additionally provides rich bounding-box annotations for each figure along with underlying numerical data. It thus offers a setting in which existing and novel visual-linguistic models can be trained from scratch and may take advantage of dense supervision. Its questions often require reference to multiple plot elements and synthesis of information distributed spatially throughout a figure. The task formulation is aimed at achieving an "intuitive" figure-understanding system, that does not resort to inverting the visualization pipeline. This is in line with the recent trend in visual-textual datasets, such as those for intuitive physics and reasoning (Goyal et al., 2017; Mun et al., 2016).

The majority of recent methods developed for VQA and related vision-language tasks, such as image captioning (Xu et al., 2015; Fang et al., 2015), video-captioning (Yu et al., 2016), phrase localization (Hu et al., 2016), and multi-modal machine translation (Elliott & Kádár, 2017), employ a neural encoder-decoder framework. These models typically encode the visual modality with pretrained CNNs, such as VGG (Simonyan & Zisserman, 2014) or ResNet (He et al., 2016), and may extract additional information from images using pretrained object detectors (Ren et al., 2015). Language encoders based on bag-of-words or LSTM approaches are typically either trained from scratch (Elliott & Kádár, 2017) or make use of pretrained word embeddings (You et al., 2016). Global or local image representations are typically combined with the language encodings through attention (Xiong et al., 2016; Yang et al., 2016; Lu et al., 2016) and pooling (Fukui et al., 2016) mechanisms, then fed to a decoder that outputs a final answer in language. In this work we evaluate a standard CNN-LSTM encoder model as well as a more recent architecture designed expressly for relational reasoning (Santoro et al., 2017).

## 3 DATASET

FigureQA consists of common scientific-style plots accompanied by questions and answers concerning them. The corpus is synthetically generated at large scale: its training set contains $100,000$ images with 1.3 million questions; the validation and test sets each contain $20,000$ images with over $250,000$ questions.

Figure 1: Sample line plot figure with question-answer pairs.

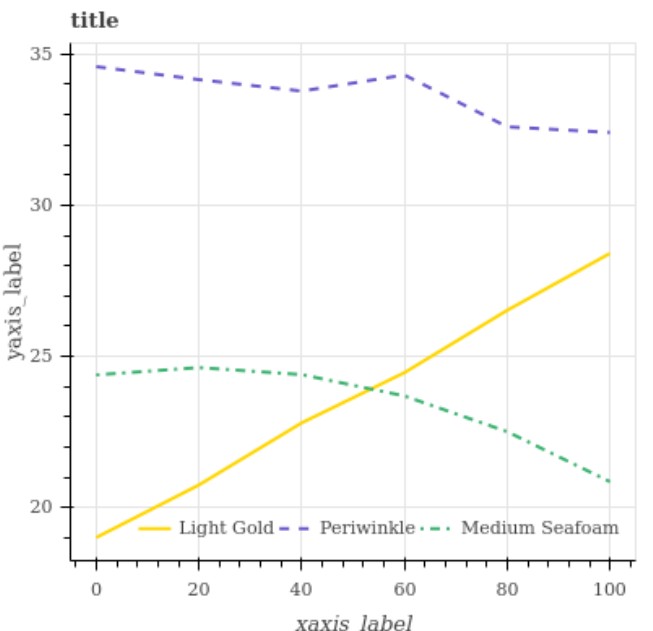

**Q:** Does Medium Seafoam intersect Light Gold?
**A: Yes**

**Q:** Is Medium Seafoam the roughest?
**A: No**

**Q:** Is Light Gold less than Periwinkle?
**A: Yes**

**Q:** Does Periwinkle have the maximum area under the curve?
**A: Yes**

**Q:** Does Medium Seafoam have the lowest value?
**A: No**

The corpus represents numerical data according to five figure types commonly found in analytical documents, namely, horizontal and vertical bar graphs, continuous and discontinuous line charts, and pie charts. These figures are produced with white background and the colors of plot elements (lines, bars and pie slices) are chosen from a set of 100 colors (see Section 3.1). Figures also contain common plot elements such as axes, gridlines, labels, and legends. We generate question-answer pairs for each figure from its numerical source data according to predefined templates. We formulate 15 questions types, given in Table 2, that compare quantitative attributes of two plot elements or one plot element versus all others. In particular, questions examine properties like the *maximum*, *minimum*, *median*, *roughness*, and *greater than/less than* relationships. All are posed as a binary choice between *yes* and *no*. In addition to the images and question-answer pairs, we provide both the source data and bounding boxes for all figure elements, and supplement questions with the names, RGB codes, and unique identifiers of the featured colors. These are for optional use in analysis or to define auxiliary training objectives.

In the following section, we describe the corpus and its generation process in depth.[2]

## 3.1 SOURCE DATA AND FIGURES

The many parameters we use to generate our source data and figures are summarized in Table 1. These constrain the data-sampling process to ensure consistent, realistic plots with a high degree of variation. Generally, we draw data values from uniform random distributions within parameter-limited ranges. We further constrain the "shape" of the data using a small set of commonly observed functions (linear, quadratic, bell curve) with additive perturbations.

A figure's data points are identified visually by color; textually (on axes and legends and in questions), we identify data points by the corresponding color names. For this purpose we chose 100 unique colors from the X11 named color set[3], selecting those with a large color distance from white (the figure background color).

We construct FigureQA's training, validation, and test sets such that all 100 colors are observed during training, while validation and testing are performed on unseen *color-plot combinations*. This

---

[2]Additional format details are available on the project website.
[3]See `https://cgit.freedesktop.org/xorg/app/rgb/tree/rgb.txt`.

we accomplish by a methodology consistent with that of the CLEVR dataset (Johnson et al., 2016), as follows. We divide our 100 colors into two disjoint, equally-sized subsets (denoted $A$ and $B$). In the training set, we color a particular figure type by drawing from one, and only one, of these subsets (see Table 1). When generating the validation and test sets, we draw from a given plot type's *opposite* subset, i.e., if subset $A$ was used for training, then subset $B$ is used for validation and testing. We refer to this as the "alternated color scheme."[4]

We define the appearance of several other aspects during data generation, randomizing these too to encourage variation. The placement of the legend within or outside the plot area is determined by a coin flip, and we select its precise location and orientation to cause minimal obstruction by counting the occupancy of cells in a $3 \times 3$ grid. Figure width is constrained to within one to two times the height, there are four font sizes available, and grid lines may be rendered or not – all with uniform probability.

Table 1: Synthetic Data Parameters, the color set column indicates the set used for training.

| Figure Types | Elements | Points | Shapes | Styles | Color Set |
|---|---|---|---|---|---|
| Vertical Bar | 1 | 2-10 | uniform random, linear, bell-shape | N/A | $A$ |
| Horizontal Bar | 1 | 2-10 | uniform random, linear, bell-shape | N/A | $B$ |
| Line | 2-7 | 5-20 | linear, linear with noise, quadratic | 5 | $A$ |
| Dot Line | 2-7 | 5-20 | linear, linear with noise, quadratic | N/A | $B$ |
| Pie | 2-7 | 1 | N/A | N/A | $A$ |

### 3.2 Questions and Answers

We generate questions and their answers by referring to a figure's source data and applying the templates given in Table 2. One *yes* and one *no* question is generated for each template that applies.

Once all question-answer pairs have been generated, we filter them to ensure an equal number of yes and no answers by discarding question-answer pairs until the answers per question type are balanced. This removes bias from the dataset to prevent models from learning summary statistics of the question-answer pairs.

Note that since we provide source data for all the figures, arbitrary additional questions may be synthesized. This makes the dataset extensible for future research.

To measure the smoothness of curves for question templates 9 and 10, we devised a roughness metric based on the sum of absolute pairwise differences of slopes, computed via finite differences. Concretely, for a curve with $n$ points defined by series $\mathbf{x}$ and $\mathbf{y}$,

$$\texttt{Roughness}(\mathbf{x}, \mathbf{y}) = \sum_{i=1}^{n-2} \left| \frac{y_{i+2} - y_{i+1}}{x_{i+2} - x_{i+1}} - \frac{y_{i+1} - y_i}{x_{i+1} - x_i} \right|.$$

### 3.3 Plotting

We generate figures from the synthesized source data using the open-source plotting library *Bokeh*. Bokeh was selected for its ease of use and modification and its expressiveness. We modified the library's web-based rendering component to extract and associate bounding boxes for all figure elements. Figures are encoded in three channels (RGB) and saved in Portable Network Graphics (PNG) format.

---

[4]We additionally provide validation and test sets built without this scheme.

[5] In the sense of *strictly greater/less* than. This clarification is provided to judges for the human baseline.

Table 2: Question Types.

| | Template | Figure Types |
|---|---|---|
| 1 | Is $X$ the minimum? | bar, pie |
| 2 | Is $X$ the maximum? | bar, pie |
| 3 | Is $X$ the low median? | bar, pie |
| 4 | Is $X$ the high median? | bar, pie |
| 5 | Is $X$ less than $Y$? | bar, pie |
| 6 | Is $X$ greater than $Y$? | bar, pie |
| 7 | Does $X$ have the minimum area under the curve? | line |
| 8 | Does $X$ have the maximum area under the curve? | line |
| 9 | Is $X$ the smoothest? | line |
| 10 | Is $X$ the roughest? | line |
| 11 | Does $X$ have the lowest value? | line |
| 12 | Does $X$ have the highest value? | line |
| 13 | Is $X$ less than $Y$?[5] | line |
| 14 | Is $X$ greater than $Y$?[5] | line |
| 15 | Does $X$ intersect $Y$? | line |

## 4 MODELS

To establish baseline performance on FigureQA, we implemented the four models described below.[6] In all experiments we use training, validation, and test sets with the alternated color scheme (see Section 3.1). The results of an experiment with the RN baseline trained and evaluated with different schemes is provided in Section C. We train all models using the *Adam* optimizer (Kingma & Ba, 2014) on the standard cross-entropy loss with learning rate 0.00025.

**Preprocessing** We resize the longer side of each image to 256 pixels, preserving the aspect ratio; images are then padded with zeros to size $256 \times 256$. For data augmentation, we use the common scheme of padding images (to size $264 \times 264$) and then randomly cropping them back to the previous size ($256 \times 256$).

**Text-only baseline** Our first baseline is a text-only model that uses an LSTM[7] to read the question word by word. Words are represented by a learned embedding of size 32 (our vocabulary size is only 85, not counting default tokens such as those marking the start and end of a sentence). The LSTM has 256 hidden units. A Multi-Layer Perceptron (MLP) classifier passes the last LSTM state through two hidden layers with 512 Rectified Linear Units (ReLUs) (Nair & Hinton, 2010) to produce an output. The second hidden layer uses dropout at a rate of 50% (Srivastava et al., 2014). This model was trained with batch size 64.

**CNN+LSTM** In this model the MLP classifier receives the concatenation of the question encoding with a learned visual representation. The visual representation comes from a CNN with five convolutional layers, each with 64 kernels of size $3 \times 3$, stride 2, zero padding of 1 on each side and batch normalization (Ioffe & Szegedy, 2015), followed by a fully connected layer of size 512. All layers use the ReLU activation function. The LSTM producing the question encoding has the same architecture as in the text-only model. This baseline was trained using four parallel workers each computing gradients on batches of size 160 which are then averaged and used for updating parameters.

**CNN+LSTM on VGG-16 features** In our third baseline we extract features from layer *pool5* of an *ImageNet*-pretrained *VGG-16* network (Simonyan & Zisserman, 2014) using the code provided with Hu et al. (2017). The extracted features (512 channels of size $10 \times 15$) are then processed by a

---

[6]The code for all baselines will be made publicly available at `https://abc.xy`.

[7]The TensorFlow (Abadi et al., 2016) implementation based on the seminal work of Hochreiter & Schmidhuber (1997).

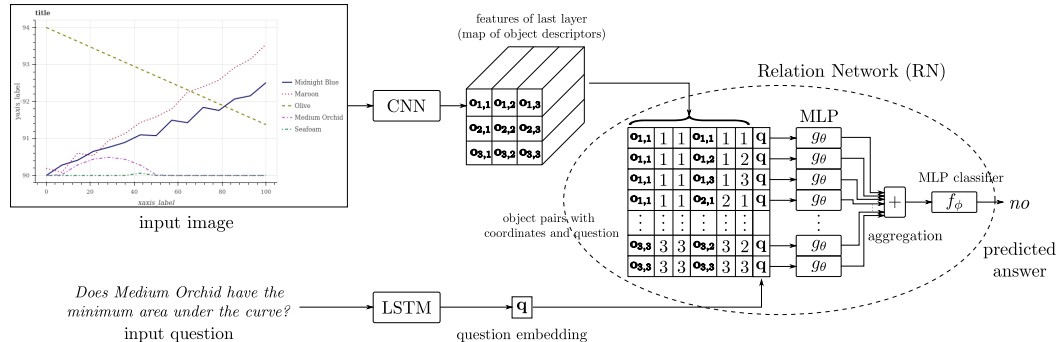

Figure 2: Sketch of the RN baseline.

CNN with four convolutional layers, all with $3 \times 3$ kernels, ReLU activation and batch normalization. The first two convolutional layers both have 128 output channels, the third and fourth 64 channels, each. The convolutional layers are followed by one fully-connected layer of size 512. This model was trained using batch size 64.

**Relation Network** Santoro et al. (2017) introduced a simple yet powerful neural module for relational reasoning. It takes as input a set of $N$ "object" representations $\mathbf{o}_i \in \mathbb{R}^C, i = 1, \dots, N$ and computes a representation of relations between objects according to

$$RN(\mathbf{O}) = f_\phi \left( \frac{1}{N^2} \sum_{i,j} g_\theta(\mathbf{o}_{i,\cdot}, \mathbf{o}_{j,\cdot}) \right), \tag{1}$$

where $\mathbf{O} \in \mathbb{R}^{N \times C}$ is the matrix containing $N$ $C$-dimensional object representations $\mathbf{o}_{i,\cdot}$ stacked row-wise. Both $f_\phi$ and $g_\theta$ are implemented as MLPs, making the relational module fully-differentiable.

In our FigureQA experiments, we follow the overall architecture used by Santoro et al. (2017) in their experiments on CLEVR from pixels, adding one convolutional layer to account for the higher resolution of our input images and increasing the number of channels. We also don't use random rotations for data augmentation, to avoid distortions that might change the correct response to a question.

The object representations are provided by a CNN with the same architecture as the one in the previous baseline, only dropping the fully-connected layer at the end. Each pixel of the CNN output (64 feature maps of size $8 \times 8$) corresponds to one "object" $\mathbf{o}_{i,\cdot} \in \mathbb{R}^{64}, i \in [1, \dots, H \cdot W]$, where $H$ and $W$, denote the height and width, respectively. To also encode the location of objects inside the feature map, the row and column coordinates are concatenated to that representation:

$$\mathbf{o}_i \leftarrow \left( o_{i,1}, \dots, o_{i,64}, \lfloor \frac{i-1}{W} \rfloor, (i-1) \,(\text{mod}\, W) \right). \tag{2}$$

The RN takes as input the stack of all pairs of object representations, concatenated with the question; here the question encoding is once again produced by an LSTM with 256 hidden units. Object pairs are then separately processed by $g_\theta$ to produce a feature representation of the relation between the corresponding objects. The sum over all relational features is then processed by $f_\phi$, yielding the predicted outputs.

The MLP implementing $g_\theta$ has four layers, each with 256 ReLU units. The MLP classifier $f_\phi$ processing the overall relational representation, has two hidden layers, each with 256 ReLU units, the second layer using dropout with a rate of $50\%$. An overall sketch of the RN's structure is shown in Figure 2. The model was trained using four parallel workers, each computing gradients on batches of size 160, which are then averaged for updating parameters.

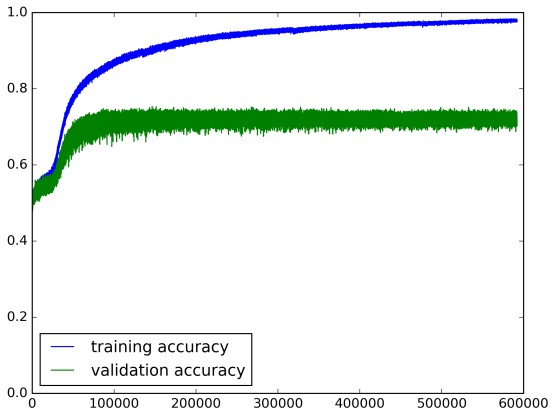

Figure 3: Learning curves of the RN.

## 5 EXPERIMENTAL RESULTS

All model baselines are trained and evaluated using the alternated color scheme. At each training step, we compute the accuracy on one randomly selected batch from validation set and keep an exponential moving average with decay 0.9. Starting from the 100th update, we perform early-stopping using the described moving average. The best performing model using this approximate validation performance measure is evaluated on the whole test set. Figure 3 shows the training and validation accuracy over updates for the RN model.

Our editorial team answered a subset from our test set, containing 16, 876 questions, corresponding to 1, 275 randomly selected figures (roughly 250 per figure type). The results are reported in Table 4 and compared with the CNN+LSTM and RN baselines evaluated on the same subset. The comparison between text only and CNN+LSTM models shows that the visual modality contributes to learning. However, due to the relational structure of the questions, the RN significantly outperforms the simpler CNN+LSTM model. Our human baseline shows that the problem is challenging, but leads by a significant margin.

Tables 5 and 6 show the performances of the CNN+LSTM and RN baselines compared to the performances of our editorial staff by figure type and by question type, respectively. More details on the human baseline and an analysis of results are provided in the supplementary material.

Table 3: Performance of our baselines on the validation and test sets with the alternated color scheme.

| Model | Validation Accuracy (%) | Test Accuracy (%) |
| --- | --- | --- |
| Text only | 50.01 | 50.01 |
| CNN+LSTM | 56.16 | 56.00 |
| CNN+LSTM on VGG-16 features | 52.31 | 52.47 |
| RN | 72.54 | 72.40 |

## 6 CONCLUSION

We introduced FigureQA, a machine learning corpus for the study of visual reasoning on scientific figures. To build this dataset, we synthesized over one million question-answer pairs grounded in

---

[8] In the sense of *strictly greater/less* than. This clarification is provided to judges for the human baseline.

Table 4: Performance of CNN+LSTM, RN and our human annotators on a subset of the test set with the alternated color scheme.

| Model | Test Accuracy (%) |
|---|---|
| CNN+LSTM | 56.04 |
| RN | 72.18 |
| Human | 91.21 |

Table 5: CNN+LSTM, RN and human accuracy (in percent) per figure type on a subset of the test set with the alternated color scheme.

| Figure Type | CNN+LSTM | RN | Human |
|---|---|---|---|
| Vertical Bar | 59.63 | 77.13 | 95.90 |
| Horizontal Bar | 57.69 | 77.02 | 96.03 |
| Line | 54.46 | 66.69 | 90.55 |
| Dot Line | 54.19 | 69.22 | 87.20 |
| Pie | 55.32 | 73.26 | 88.26 |

over $100,000$ synthetic figure images. Questions examine plot characteristics like the extrema, area-under-the-curve, smoothness, and intersection, and require integration of information distributed spatially throughout a figure. The corpus comes bundled with side data to facilitate the training of machine learning systems. This includes the numerical data used to generate each figure and bounding-box annotations for all plot elements. We studied the visual reasoning task by training four baseline neural models on our data, analyzing their test-set performance, and comparing it with that of humans. Results indicate that more powerful models must be developed to reach human-level performance.

In future work, we plan to test the transfer of models trained on FigureQA to question-answering on real scientific figures, and to iteratively extend the dataset either by significantly increasing the number of templates or by crowdsourcing natural-language questions. We envision FigureQA as a first step to developing models that intuitively extract knowledge from the numerous figures produced by modern science.

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

Table 6: CNN+LSTM, RN and human accuracy (in percent) per question type. The reported accuracies are both computed on the same subset of the test set with alternated color scheme.

|   | Template | CNN+LSTM | RN | Human |
|---|----------|----------|-----|-------|
| 1 | Is $X$ the minimum? | 56.63 | 76.78 | 97.06 |
| 2 | Is $X$ the maximum? | 58.54 | 83.47 | 97.18 |
| 3 | Is $X$ the low median? | 53.66 | 66.69 | 86.39 |
| 4 | Is $X$ the high median? | 53.53 | 66.50 | 86.91 |
| 5 | Is $X$ less than $Y$? | 61.36 | 80.49 | 96.15 |
| 6 | Is $X$ greater than $Y$? | 61.23 | 81.00 | 96.15 |
| 7 | Does $X$ have the minimum area under the curve? | 56.60 | 69.57 | 94.22 |
| 8 | Does $X$ have the maximum area under the curve? | 55.69 | 78.45 | 95.36 |
| 9 | Is $X$ the smoothest? | 55.49 | 58.57 | 78.02 |
| 10 | Is $X$ the roughest? | 54.52 | 56.28 | 79.52 |
| 11 | Does $X$ have the lowest value? | 55.08 | 69.65 | 90.33 |
| 12 | Does $X$ have the highest value? | 58.90 | 76.23 | 93.11 |
| 13 | Is $X$ less than $Y$?[8] | 50.62 | 67.75 | 90.12 |
| 14 | Is $X$ greater than $Y$?[8] | 51.00 | 67.12 | 89.88 |
| 15 | Does $X$ intersect $Y$? | 49.88 | 68.75 | 89.62 |

Hao Fang, Saurabh Gupta, Forrest Iandola, Rupesh K Srivastava, Li Deng, Piotr Dollár, Jianfeng Gao, Xiaodong He, Margaret Mitchell, John C Platt, et al. From captions to visual concepts and back. In *Proceedings of the IEEE conference on computer vision and pattern recognition*, pp. 1473–1482, 2015.

Akira Fukui, Dong Huk Park, Daylen Yang, Anna Rohrbach, Trevor Darrell, and Marcus Rohrbach. Multimodal compact bilinear pooling for visual question answering and visual grounding. *arXiv preprint arXiv:1606.01847*, 2016.

Kunihiko Fukushima. Neocognitron: A hierarchical neural network capable of visual pattern recognition. *Neural networks*, 1(2):119–130, 1988.

Raghav Goyal, Samira Kahou, Vincent Michalski, Joanna Materzyńska, Susanne Westphal, Heuna Kim, Valentin Haenel, Ingo Fruend, Peter Yianilos, Moritz Mueller-Freitag, et al. The" something something" video database for learning and evaluating visual common sense. *arXiv preprint arXiv:1706.04261*, 2017.

Yash Goyal, Tejas Khot, Douglas Summers-Stay, Dhruv Batra, and Devi Parikh. Making the v in vqa matter: Elevating the role of image understanding in visual question answering. *arXiv preprint arXiv:1612.00837*, 2016.

Kaiming He, Xiangyu Zhang, Shaoqing Ren, and Jian Sun. Deep residual learning for image recognition. In *Proceedings of the IEEE conference on computer vision and pattern recognition*, pp. 770–778, 2016.

Sepp Hochreiter and Jürgen Schmidhuber. Long short-term memory. *Neural computation*, 9(8): 1735–1780, 1997.

Ronghang Hu, Huazhe Xu, Marcus Rohrbach, Jiashi Feng, Kate Saenko, and Trevor Darrell. Natural language object retrieval. In *Proceedings of the IEEE Conference on Computer Vision and Pattern Recognition*, pp. 4555–4564, 2016.

Ronghang Hu, Jacob Andreas, Marcus Rohrbach, Trevor Darrell, and Kate Saenko. Learning to reason: End-to-end module networks for visual question answering. *arXiv preprint arXiv:1704.05526*, 2017.

Sergey Ioffe and Christian Szegedy. Batch normalization: Accelerating deep network training by reducing internal covariate shift. In *International Conference on Machine Learning*, pp. 448–456, 2015.

Justin Johnson, Bharath Hariharan, Laurens van der Maaten, Li Fei-Fei, C Lawrence Zitnick, and Ross Girshick. Clevr: A diagnostic dataset for compositional language and elementary visual reasoning. *arXiv preprint arXiv:1612.06890*, 2016.

Daekyoung Jung, Wonjae Kim, Hyunjoo Song, Jeong-in Hwang, Bongshin Lee, Bohyoung Kim, and Jinwook Seo. Chartsense: Interactive data extraction from chart images. In *Proceedings of the 2017 CHI Conference on Human Factors in Computing Systems*, pp. 6706–6717. ACM, 2017.

Diederik Kingma and Jimmy Ba. Adam: A method for stochastic optimization. *arXiv preprint arXiv:1412.6980*, 2014.

Yann LeCun, Léon Bottou, Yoshua Bengio, and Patrick Haffner. Gradient-based learning applied to document recognition. *Proceedings of the IEEE*, 86(11):2278–2324, 1998.

Jiasen Lu, Jianwei Yang, Dhruv Batra, and Devi Parikh. Hierarchical question-image co-attention for visual question answering. In *Advances In Neural Information Processing Systems*, pp. 289–297, 2016.

Jonghwan Mun, Paul Hongsuck Seo, Ilchae Jung, and Bohyung Han. Marioqa: Answering questions by watching gameplay videos. *arXiv preprint arXiv:1612.01669*, 2016.

Vinod Nair and Geoffrey E Hinton. Rectified linear units improve restricted boltzmann machines. In *Proceedings of the 27th international conference on machine learning (ICML-10)*, pp. 807–814, 2010.

Ethan Perez, Harm de Vries, Florian Strub, Vincent Dumoulin, and Aaron Courville. Learning visual reasoning without strong priors. *arXiv preprint arXiv:1707.03017*, 2017.

Jorge Poco and Jeffrey Heer. Reverse-engineering visualizations: Recovering visual encodings from chart images. In *Computer Graphics Forum*, volume 36, pp. 353–363. Wiley Online Library, 2017.

Shaoqing Ren, Kaiming He, Ross Girshick, and Jian Sun. Faster R-CNN: Towards real-time object detection with region proposal networks. *arXiv preprint arXiv:1506.01497*, 2015.

Adam Santoro, David Raposo, David GT Barrett, Mateusz Malinowski, Razvan Pascanu, Peter Battaglia, and Timothy Lillicrap. A simple neural network module for relational reasoning. *arXiv preprint arXiv:1706.01427*, 2017.

Noah Siegel, Zachary Horvitz, Roie Levin, Santosh Divvala, and Ali Farhadi. Figureseer: Parsing result-figures in research papers. In *European Conference on Computer Vision*, pp. 664–680. Springer, 2016.

Karen Simonyan and Andrew Zisserman. Very deep convolutional networks for large-scale image recognition. *arXiv preprint arXiv:1409.1556*, 2014.

Nitish Srivastava, Geoffrey E Hinton, Alex Krizhevsky, Ilya Sutskever, and Ruslan Salakhutdinov. Dropout: a simple way to prevent neural networks from overfitting. *Journal of machine learning research*, 15(1):1929–1958, 2014.

Alane Suhr, Mike Lewis, James Yeh, and Yoav Artzi. A corpus of natural language for visual reasoning. In *55th Annual Meeting of the Association for Computational Linguistics, ACL*, 2017.

Satoshi Tsutsui and David Crandall. A data driven approach for compound figure separation using convolutional neural networks. *arXiv preprint arXiv:1703.05105*, 2017.

Caiming Xiong, Stephen Merity, and Richard Socher. Dynamic memory networks for visual and textual question answering. In *International Conference on Machine Learning*, pp. 2397–2406, 2016.

Kelvin Xu, Jimmy Ba, Ryan Kiros, Kyunghyun Cho, Aaron Courville, Ruslan Salakhudinov, Rich Zemel, and Yoshua Bengio. Show, attend and tell: Neural image caption generation with visual attention. In *International Conference on Machine Learning*, pp. 2048–2057, 2015.

Zichao Yang, Xiaodong He, Jianfeng Gao, Li Deng, and Alex Smola. Stacked attention networks for image question answering. In *Proceedings of the IEEE Conference on Computer Vision and Pattern Recognition*, pp. 21–29, 2016.

Quanzeng You, Hailin Jin, Zhaowen Wang, Chen Fang, and Jiebo Luo. Image captioning with semantic attention. In *Proceedings of the IEEE Conference on Computer Vision and Pattern Recognition*, pp. 4651–4659, 2016.

Haonan Yu, Jiang Wang, Zhiheng Huang, Yi Yang, and Wei Xu. Video paragraph captioning using hierarchical recurrent neural networks. In *Proceedings of the IEEE conference on computer vision and pattern recognition*, pp. 4584–4593, 2016.

## A    DATA SAMPLES

Here we present a sample figures of each plot type (*vertical bar graph*, *horizontal bar graph*, *line graph*, *dot line graph* and *pie chart*) from our dataset along with the corresponding question-answer pairs and some of the bounding boxes.

### A.1    VERTICAL BAR GRAPH

Figure 4: Vertical bar graph with question answer pairs.

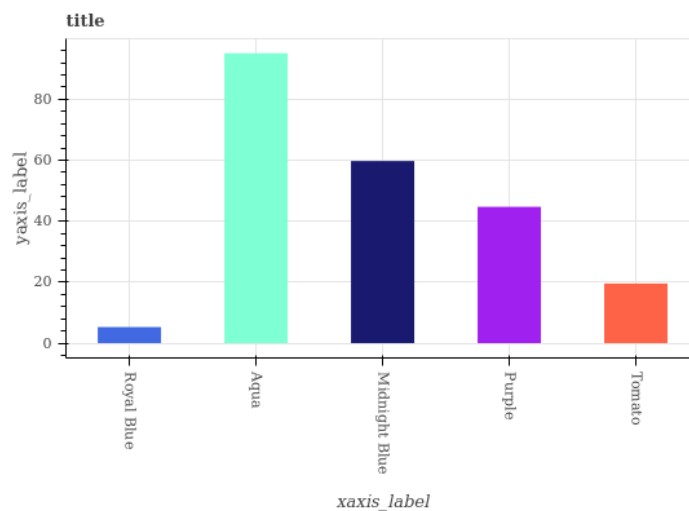

**Q:** Is Aqua the maximum?
**A: Yes**

**Q:** Is Midnight Blue greater than Aqua?
**A: No**

**Q:** Is Midnight Blue less than Aqua?
**A: Yes**

**Q:** Is Purple the high median?
**A: Yes**

**Q:** Is Tomato the low median?
**A: No**

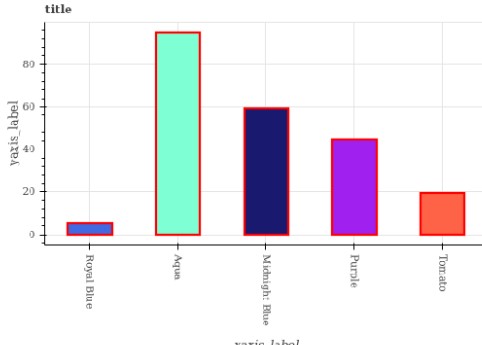

Figure 5: Vertical bar graph with some annotations.

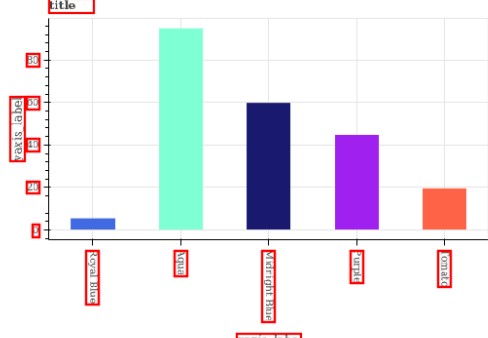

Figure 6: Vertical bar graph with label annotations.

## A.2 HORIZONTAL BAR GRAPH

Figure 7: Horizontal bar graph with question answer pairs.

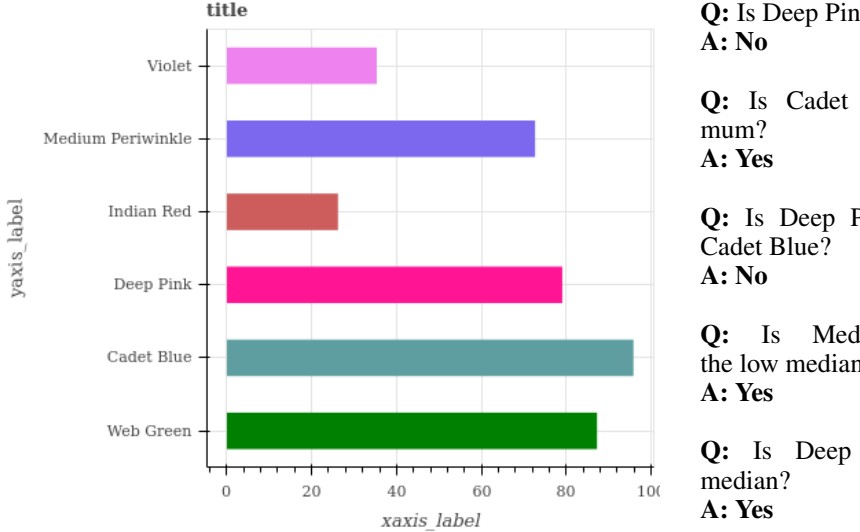

**Q:** Is Deep Pink the minimum?
**A: No**

**Q:** Is Cadet Blue the maximum?
**A: Yes**

**Q:** Is Deep Pink greater than Cadet Blue?
**A: No**

**Q:** Is Medium Periwinkle the low median?
**A: Yes**

**Q:** Is Deep Pink the high median?
**A: Yes**

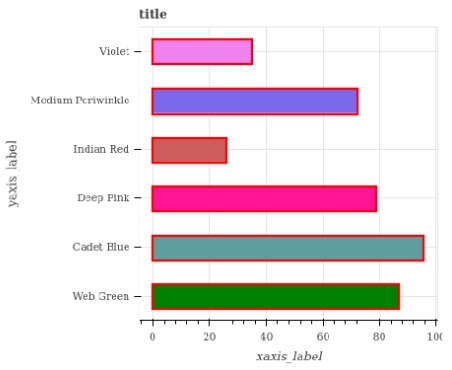

Figure 8: Horizontal bar graph with some annotations.

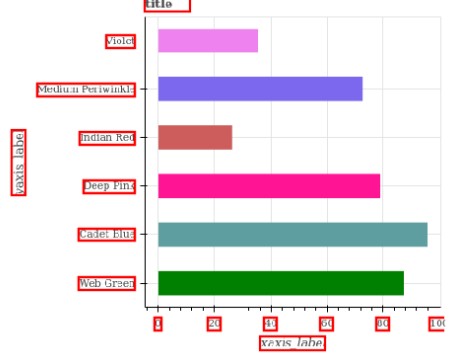

Figure 9: Horizontal bar graph with label annotations.

## A.3    LINE GRAPH

Figure 10: Line graph with question answer pairs.

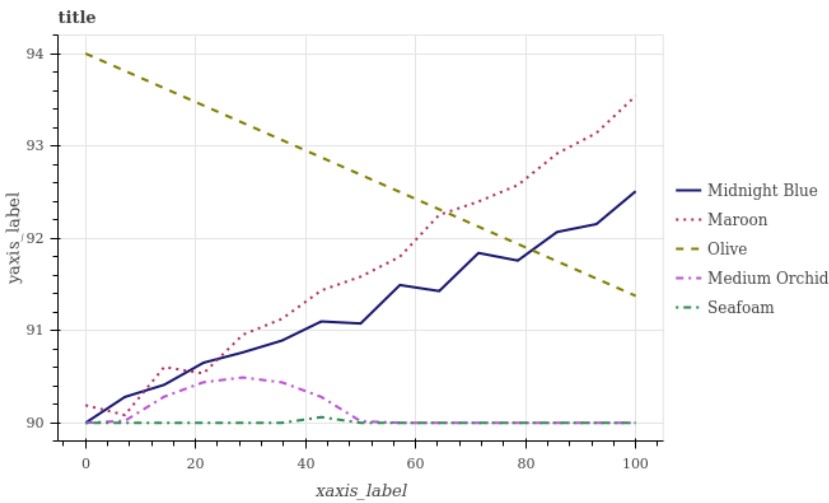

**Q:** Does Medium Orchid have the minimum area under the curve?
**A: No**

**Q:** Is Olive the smoothest?
**A: Yes**

**Q:** Does Olive have the highest value?
**A: Yes**

**Q:** Is Seafoam less than Olive?
**A: Yes**

**Q:** Does Olive intersect Midnight Blue?
**A: Yes**

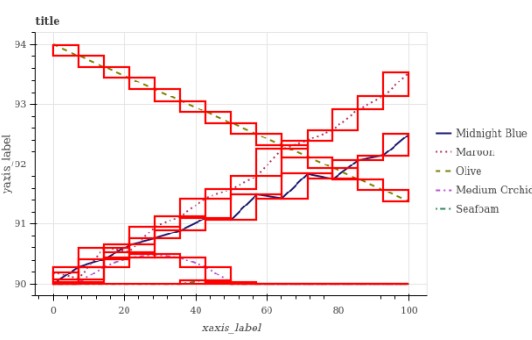

Figure 11:    Line  graph  with  some annotations.

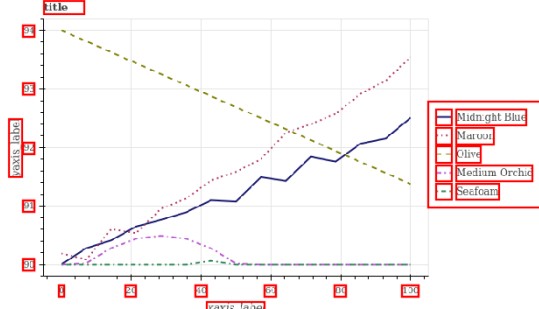

Figure 12:    Line  graph  with  label annotations.

## A.4    DOT LINE GRAPH

Figure 13: Dot line graph with question answer pairs.

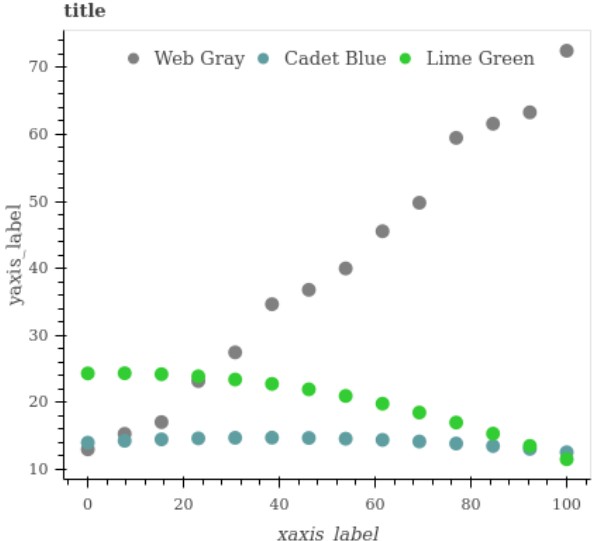

**Q:** Does Web Gray have the maximum area under the curve?
**A: Yes**

**Q:** Does Cadet Blue have the minimum area under the curve?
**A: Yes**

**Q:** Is Web Gray the roughest?
**A: Yes**

**Q:** Does Lime Green have the lowest value?
**A: Yes**

**Q:** Is Lime Green less than Web Gray?
**A: No**

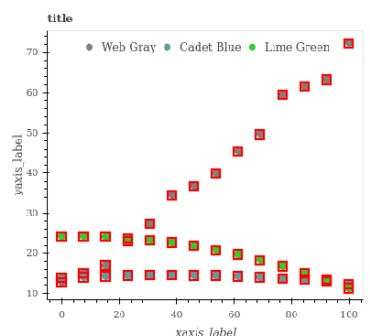

Figure 14: Dot line graph with some annotations.

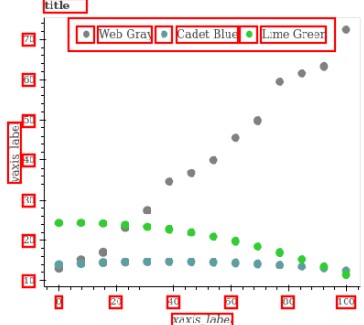

Figure 15: Dot line graph with label annotations.

### A.5   PIE CHART

Figure 16: Pie chart with question answer pairs.

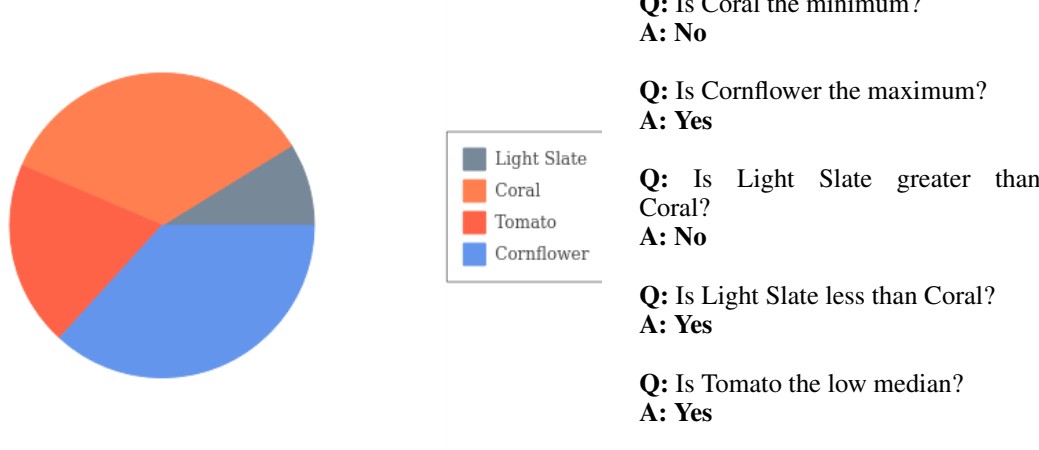

**Q:** Is Coral the minimum?
**A: No**

**Q:** Is Cornflower the maximum?
**A: Yes**

**Q:** Is Light Slate greater than Coral?
**A: No**

**Q:** Is Light Slate less than Coral?
**A: Yes**

**Q:** Is Tomato the low median?
**A: Yes**

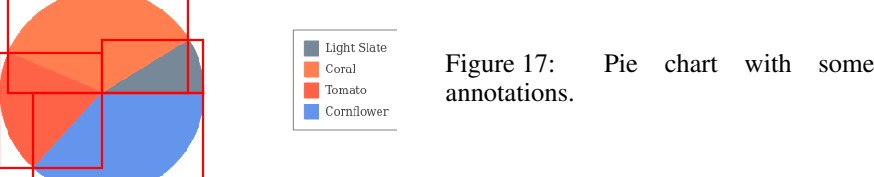

Figure 17:   Pie   chart   with   some annotations.

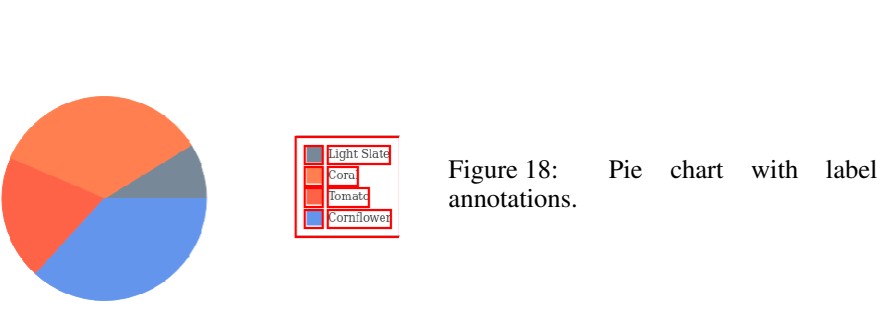

Figure 18:   Pie   chart   with   label annotations.

## B  HUMAN BASELINE

To assess FigureQA's difficulty and to set a benchmark for model performance, we measured human accuracy on a sample of the test set with the alternated color scheme. Our editorial staff answered $16,876$ questions corresponding to $1,275$ randomly selected figures (roughly 250 per type), providing them in each instance with a figure image, a question, and some disambiguation guidelines. Our editors achieved an accuracy of 91.21%, compared with 72.18% for the RN Santoro et al. (2017) baseline. We provide further analysis of the human results below.

### B.1  PERFORMANCE BY FIGURE TYPE

We stratify human accuracy by figure type in Table 5. People performed exceptionally well on bar graphs, though worse on line plots, dot-line plots, and pie charts. Analyzing the results and plot images from these figure categories, we learned that pie charts with similarly sized slices led most frequently to mistakes. Accuracy on dot-line plots was lower because plot elements sometimes obscure each other as Figure 21 shows.

Figure 19: Sample pie chart with visually ambiguous attributes. The Sandy Brown, Web Gray, and Tan slices all have similar arc length.

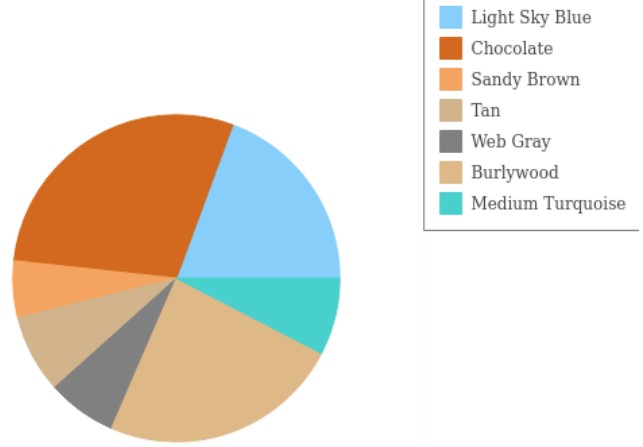

### B.2  PERFORMANCE BY QUESTION TYPE

Table 6 shows how human accuracy varies across question types, with people performing best on minimum, maximum, and greater/less than queries. Accuracy is generally higher on question types for categorical figures compared to continuous figures. It is noticeably lower for questions concerning the median and curve smoothness. Analysis indicates that many wrong answers to median questions occurred when plots had a larger number of (unordered) elements, which increases the difficulty of the task and may also induce an optical illusion. In the case of smoothness, annotators struggled to consider both the number of deviations in a curve and the size of deviations. This was particularly evident when comparing one line with more deviations to another with larger ones. Additionally, ground truth answers for smoothness were determined with computational or numerical precision that is beyond the capacity of human annotators. In some images, smoothness differences were too small to notice accurately with the naked eye.

### B.3  UNKNOWN ANSWERS

We provided our annotators with a third answer option, *unknown*, for cases where it was difficult or impossible to answer a question unambiguously. Note that we instructed our annotators to select *unknown* as a last resort. Only **0.34%** of test questions were answered with *unknown*, and this accounted for **3.91%** of all incorrect answers. Looking at the small number of such responses, we observe that generally, annotators selected *unknown* in cases where two colors were difficult to

Figure 20: Sample figures with wrong answers illustrating common issues per question type.

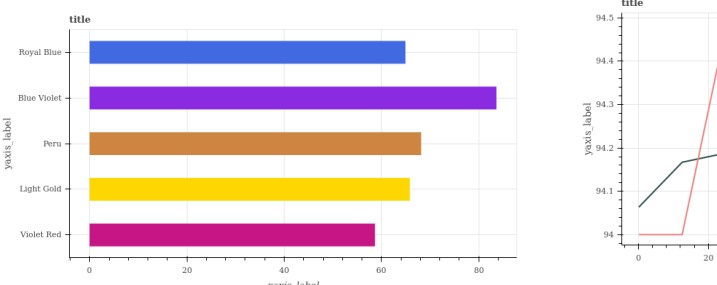
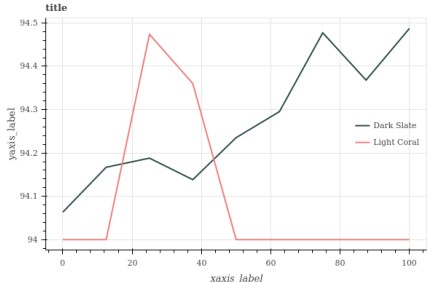

Which bar is the median: Light Gold or Royal Blue?

Which curve is rougher? One seems 'noisier' while another seems more 'jagged'.

distinguish from each other, when one plot element was covered by another, or when a line plot's region of interest was obscured by a legend.

Figure 21: Sample figures with unknown answers provided by human annotators.

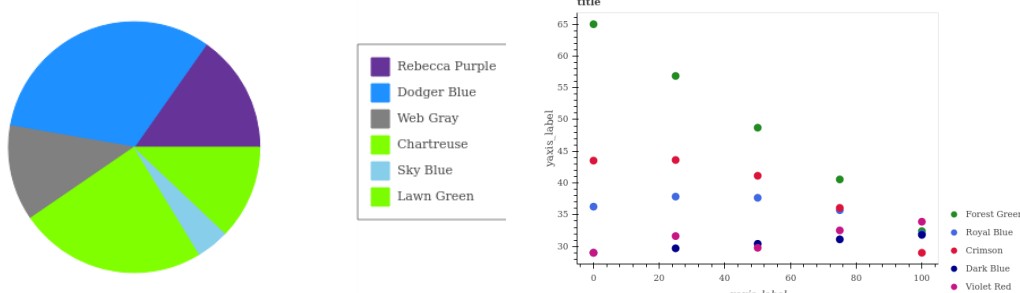

Q: Is Chartreuse the high median?

Q: Does Dark Blue intersect Royal Blue?

## C  PERFORMANCE OF THE RELATION NETWORK WITH AND WITHOUT ALTERNATED COLOR SCHEME

In this experiment we trained the RN baseline using both the validation set with swapped colors and without, saving parameters for both. We then evaluated both models on the test sets with and without swapped colors. Table 7 compares the results.

Table 7: Performance of our RN baselines trained with early stopping on *val1* (using the same color-to-plot-type assignments as in the training set) and with early stopping on *val2* (with swapped colors). We show performances of both on *test1* (with the same color assignments) and *test2* (with swapped colors).

| Model | test1 Accuracy (%) | test2 Accuracy (%) |
|---|---|---|
| RN (val1) | 67.74 | 60.35 |
| RN (val2) | 76.52 | 72.40 |

