# OpenReview forum: "FigureQA: An Annotated Figure Dataset for Visual Reasoning"
_ICLR.cc/2018/Conference — Invite to Workshop Track_

### Official Review · AnonReviewer2 · 2017-11-27
**Review from AnonReviewer2**

**Rating:** 6
**Confidence:** 4

**Review:**

This paper introduces a new dataset called FigureQA. The images are synthetic scientific style figures and questions are generated from 15 templates which concern various relationships between plot elements. The authors experiment with the proposed dataset with 3 baselines. Text-only baseline, which only considers the questions; CNN+LSTM baseline, which does a late fusion between image and question representation; Relation Network, which follows Santoro et al. (2017). Experiment results show that the proposed task poses a difficult challenge where CNN+LSTM baseline only 2% better than guessing (50%) and relation network which takes spatial reasoning into consideration, performs better on this task (61.54%).

[Strenghts]

The proposed Figure QA dataset is a first step towards developing models that can recognize the visual representation of data. This is definitely a novel area that requires the machine not only understand the corpus, but also the scientific figure associated with the figure. Traditional VQA methods not working well on the proposed dataset, only 2% better than guessing (50%). The proposed dataset requires the machine understand the spatial arrangement of the object and reason the relations between them.

[Weaknesses]

1: There are no novel algorithms associated with this dataset. CVPR seems a better place to publish this paper, but I'm open to ICLR accept dataset paper.

2: The generated templated questions of proposed FigureQA dataset is very constraint.  All the question is the binary question, and there is no variation of the template with respect to the same question type. Most of the question type can be represented as a triplet. In this sense, the proposed dataset requires less language understanding compare to previous synthesis dataset such as CLEVER.

3: Since the generated question is very templated and less variational, a traditional hand-crafted approach may perform much better compared to the end-to-end approach. The paper didn't have any baseline for the hand-crafted approach, thus we don't know how it performs on the proposed dataset, and whether more advanced neural approaches are needed for this dataset.

[Summary]

This paper introduces a new dataset called FigureQA, which answering the synthetic scientific style figures given the 15 type of questions. The authors experiment the proposed dataset with 3 neural baselines: Question only, LSTM + CNN and relational network. The proposed dataset is the first step towards developing models that can recognize the visual representation of data. However, my major concern about this dataset is the synthesized question is very templated and less variational. From the example, it seems it does not require natural language understanding of the dataset. Triplet representation seems enough to represent the most type of the question. The authors also didn't conduct hand-crafted baselines, which can provide a better intuition about the difficulty lies in the proposed dataset. Taking all these into account, I suggest accepting this paper if the authors could provide more justification on the question side of the proposed dataset.

---

> ### Author Response · Authors · 2018-01-05
> **Rebuttal (issues raised by Reviewer 2)**
>
> Thank you very much for your review and the constructive criticism.
> Please find below our responses (each below a brief summary of the corresponding issue):
>
> 1) No novel algorithms associated with data set. CVPR would be better
> --> The manuscript is a dataset paper, that focuses on motivating and introducing a novel task and providing baseline models for benchmarking. We are of the opinion that dataset papers fit well into a conference on representation learning, as long as they target weaknesses of current representation learning algorithms.
> Regarding the choice of venue: We were split between submitting to ICLR or CVPR and decided to submit to ICLR, due to potential restrictions for travelling to the US.
>
> 2) Very constraint question templates, all are binary questions, no variation of language w.r.t. the same question type. Could use triplet representation instead of LSTM.
> --> In early experiments we did just that as we thought the same, but the LSTM still had a slight edge over those experiments. As there is not too much overhead, due to relatively short questions and we plan to extend the corpus, we just kept the LSTM. In future versions we plan to add more natural language variation, either by significantly increasing the number of templates based on feedback from the community or via crowdsourcing. We are collecting candidate templates for the next version, but as the experiments show, the current task already poses a challenge.
> We would also like to mention that the additional datapoint annotations allows the dataset to be extended to any type of question or answer.
>
> 3) A handcrafted approach might perform better
> --> Thank you for the suggestion. One could surely engineer an approach that might perform better, but the pipelines would probably differ between plot types. Also one would probably have to encode stronger prior knowledge in the features beyond the priors introduced by using a convolutional network or a relational network. Representation learning allows the model to learn to deal with all plot types in the training set.
> We have added a baseline using VGG (pretrained on ImageNet) features as input, which does not perform well, suggesting that either pretraining representations on a more related data set or end-to-end training might be necessary.
> We are not aware of any existing specific handcrafted approaches, that we could have evaluated on FigureQA without major changes to the code base.
>
> Concerning your question earlier about validation scores: We have added them to the revised manuscript. We also updated the implementation of the RN and CNN+LSTM baselines to use multiple GPUs and added the new improved results to the manuscript.
> We hope that our revisions and the rebuttal address all of your concerns.

---

> > ### Comment · AnonReviewer2 · 2018-01-16
> > **Final recommendation**
> >
> > After reading the rebuttal, I'm satisfied with the response in terms of val and test split performance. However, the question part of the proposed dataset is automatically generated without any variations. This is my major concerns about this paper. Considering this paper is the first step towards research in the visual representation of data. I'll improve my rating from 4 to 6.

---

### Official Review · AnonReviewer1 · 2017-11-27
**Task motivation, experiments on FigureSeer, experiments with quantitative data and bounding box annotations.**

**Rating:** 6
**Confidence:** 3

**Review:**

Summary:
The paper introduces a new dataset (FigureQA) of question answering on figures (line plots, bar graphs, pie charts). The questions involve reasoning about figure elements (e.g., “Is X minimum?”, “Does X have maximum area under curve?”, where X refers to an element in a figure). The images are synthetic and the questions are templated. The dataset consists of over 100,000 images and the questions are from 15 templates. The authors also provide the numerical data associated with each figure and bounding box annotations for all plot elements. The paper trains and evaluates three baselines – question only LSTM, CNN + LSTM and Relation Networks on the proposed FigureQA dataset. The experimental results show that Relation Networks outperform the other two baselines, but still ~30% behind human performance.

Strengths:
1.	The proposed task is a useful task for building intelligent AI agents.
2.	The writing of the paper is clear with enough details about the data generation process.
3.	The idea of using different compositions of color and figure during train and test is interesting.
4.	The baselines experimented with in the paper make sense.

Weaknesses:
1.	The motivation behind the proposed task needs to be better elaborated. As of now, the paper mentions in one line that automatic understanding of figures could help human analysists. But, it would be good if this can be supported with real life examples.
2.	The dataset proposed in the paper comes with bounding box annotations, however, the usage of bounding boxes isn’t clear. The paper briefly mentions supervising attention models using such boxes, but it isn’t clear how bounding boxes for data points could be used.
3.	It would have been good if the paper had the experiments on reconstructing quantitave data from plots and using bounding boxes for providing attention supervision, in order to concretize the usage of these annotations?
4.	The paper mentions the that analyzing the performance of the models trained on FigureQA on real datasets would help extend the FigureQA corupus. So, why didn’t authors try the baselines models on FigureSeer dataset?
5.	It is not clear why did the authors devise a new metric for smoothness and not use existing metrics?
6.	The paper should clarify which CNN is used for CNN + LSTM and Relation Networks models?
7.	The paper should clarify the loss function used to train the models? Is it binary cross entropy loss?
8.	The paper does not mention the accuracy using the quantitative data associated with the plot? Is it 100%? What if two quantities being questions about are equal and the question is about finding the max/min? How much is the error due to such situations?

---

> ### Author Response · Authors · 2018-01-05
> **Rebuttal (issues raised by Reviewer 1)**
>
> Thank you very much for your review and the constructive criticism.
> Please find below our responses (each below a brief summary of the corresponding issue):
>
> 1) Elaborate motivation of the proposed task. The manuscript only mentions that automatic understanding of figures could help human analysts. Real-life examples would be good.
> --> a) One example in real life is that people who work in finance have to interpret large amounts of simple plots everyday. A computer vision algorithm that can assist here, could save time.
> Apart from financial data and plots, the skill of interpreting figures and understanding visual information is very useful and plays a role in education. Similar questions to those we have in FigureQA (among more complicated ones) are usually part of Graduate Record Examinations (GRE). Our choice of plot types and question types were in part motivated by such exams (see for example here: https://www.ets.org/s/gre/accessible/gre_practice_test_3_quant_18_point.pdf pages: 29, 44, 45, 86). We have added/emphasized this point in the revised manuscript.
>
> 2) The manuscript mentions that bounding boxes could be used for supervising attention models, but doesn’t clearly describe how.
> --> One example would be to use an attention window to extract image patches containing one or multiple objects of interest (e.g. line segments in a line plot). We initially thought this might be necessary to get models to train at all, but found that training without auxiliary objectives achieved a performance significantly above chance. Because a significant amount of work went into extracting bounding boxes using a modified version of Bokeh, we decided to include them in the release, in case someone wants to use the data for a different task, such as bounding box prediction or reconstruction of data coordinates. The revised manuscript now clarifies this and clearly states that experiments using attention models or extraction of coordinates are outside of the scope of this work.
>
> 3) Why no experiments on reconstruction of quantitative data or attention?
> --> This is partially answered under 2). But in addition, the focus of our work is on intuitive understanding of figures based on visual cues, rather than inversion of the visualization pipeline. Humans usually don’t build a table of coordinates to reach the conclusion that one of the curves seems smoother or that a bar is greater than another.
> The revised manuscript puts more emphasis on our focus on intuitive figure understanding.
>
> 4) The manuscript mentions that analyzing performance of models trained on FigureQA on real data could help to extend the FigureQA dataset. Why didn’t the authors add such experiments on FigureSeer?
> -->This is a good point, and we can elaborate more on our decision to not use the FigureSeer dataset (Siegel et al., 2016).
> The FigureSeer paper claims to have 3,500 QA pairs for a subset of the figures, but these types of questions and answer types are not found in our dataset. The questions are templated, concerning a dataset (for figure retrieval) and a metric (for figure analysis), for which the answer is the numerical value of the metric in that dataset figure. The metrics are not consistent with the question types in FigureQA and the answers are non-binary, so we could not train our baselines on that data. The FigureSeer questions were also not publicly available and were not provided when we requested the full version of the dataset.
> Data point annotations included for a subset of 1,000 FigureSeer images were crowdsourced and aren’t reliable for generating figure images or question-answer pairs. Figures with many points are often missing points (e.g. 01951-10.1.1.20.2491-Figure-1 from the dataset sample available here: http://ai2-website.s3.amazonaws.com/data/FigureSeerDataset.zip). These 1,000 annotated images are all line plots, which only covers one fifth of the figure types available in FigureQA, so they are only a starting point.
> Finally, annotating a portion of the FigureSeer dataset as a real-world test set was infeasible given the limited time we had to prepare this rebuttal, though we intend to complete this for the next version of the dataset.
> [we had to split the rebuttal into two parts, this is part 1]

---

> > ### Author Response · Authors · 2018-01-05
> > **Rebuttal (issues raised by Reviewer 1, continued)**
> >
> > 5) Why a new metric for smoothness?
> > --> We felt that second-order derivatives across the curve would be sufficient, as the magnitude of the second-order derivative correlates to “bumpiness” as perceived by humans. We approximate the second-order derivative by finite differences in the curve itself. Our roughness measure is similar to the second derivative calculation for a quadratic interpolant with Lagrangian basis polynomials (see Equation 15.44 in https://www.rsmas.miami.edu/users/miskandarani/Courses/MSC321/lectfiniteDifference.pdf
> > We decided against using a surface roughness measure or variance because these measures do not work well for globally “rough” curves, like quadratics, due to their high deviation from the mean of the line plot.
> > One alternative roughness measure would have been lag-X autocorrelation, though experimentation is necessary to find the right lag parameter and we felt it would be better to have an objective, parameter-free model.
> >
> > 6) The paper should clarify which CNN is used for CNN + LSTM and Relation Network
> > --> We completely specified the CNNs both for the CNN + LSTM and the RN in the initial draft of the manuscript. All hyperparameters required for implementation can be found in the respective paragraphs in Section 4 (“Models”).
> > The paragraph “CNN+LSTM” says:
> > “The visual representation comes from a CNN with five convolutional layers, each with 64 kernels of size $3\times3$, stride 2, zero padding of 1 on each side and batch normalization (Ioffe et al. 2015), followed by a fully connected layer of size 512. All layers use the \gls{relu} activation function.”
> > And the paragraph of the RN mentions that the same CNN architecture without the fully-connected layer is used.
> >
> > 7) The paper should clarify which loss function is used. Is it the binary cross-entropy?
> > --> Yes, we used the standard binary cross-entropy loss. Thanks for pointing this out. We added this information in the revised manuscript.
> >
> > 8) The paper does not mention the accuracies using quantitative data. How much error is due to min/max questions with two equal quantities?
> > --> Quantitative data is not used in any of our models, as we are aiming to achieve intuitive visual understanding of figures.
> > If two quantities are the same and greater than all others, then both are maxima. Because of potential misunderstandings (is the maximum meant to be unique?) we avoided this special case in data generation.

---

> > > ### Comment · AnonReviewer1 · 2018-01-13
> > > **Post-rebuttal evaluation**
> > >
> > > After reading the authors' responses to the concerns raised in the review by me and my fellow reviewers, I would like to recommend acceptance of this paper because it proposes a novel task which seems useful for building intelligent AI agents and the dataset proposed in the paper is a good starting point.

---

### Official Review · AnonReviewer3 · 2017-11-29
**Poorly motivated and needs more analysis**

**Rating:** 6
**Confidence:** 4

**Review:**

Summary:
The paper introduces a new visual reasoning dataset called Figure-QA which consists of 140K figure images and 1.55M QA pairs. The images are generated synthetically by plotting perturbed sampled data using a visualization tool. The questions are also generated synthetically using 15 templates. Performance of baseline models and humans show that it is a challenging task and more advanced models are required to solve this task.

Strengths:
— FigureQA can help in developing models that can extract useful information from visual representations of data.
— Since performance on CLEVR dataset is already close to 100%, more challenging visual reasoning datasets would encourage the community to develop more advanced reasoning models. One of such datasets can be FigureQA.
— The paper is well written and easy to follow.


Weaknesses:
— Since the dataset is created synthetically, it is not clear if it is actually visual reasoning which is needed to solve this task, or the models can exploit biases (not necessarily language biases) to perform well on this dataset. In short, how do we know if the models trained on this dataset are actually learning something useful? One way to ensure this would be to show that models trained on this dataset can perform well on some other task. The first thing to try to show the usefulness of FigureQA is to show that the models trained on FigureQA dataset perform well on a real (figure, QA) dataset.
— The only advantages mentioned in the paper of using a synthetic dataset for this task are having greater control over task’s complexity and enabling auxiliary supervision signals, but none of them are shown in this paper, so it’s not clear if they are needed or useful.
— The paper should discuss what type of abilities are required in the models to perform well on this task, and how these abilities are currently not studied in the research community. Or in short, what new challenges are being introduced by FigureQA and how should researchers go about solving them on a high level?
— With what goal were these 15 types of questions chosen? Are these the most useful questions analysts want to extract out of plots? I am especially concerned about finding the roughest/smoothest and low/high median. Even humans are relatively bad at these tasks. Why do we expect models to do well on them?
— Why only binary questions? It is probably more difficult for analysts to ask a binary question than to ask non-binary ones such as “What is the highest in this plot?” — Why these 5 types of plots? Can the authors justify that these 5 types of plots are the most frequent ones dealt by analysts?
— Are the model accuracies in Table 3 on the same subset as humans or on the complete test set? Can the authors please report both separately?


Overall:
The proposed dataset seems reasonable but neither the dataset seems properly motivated (something where analysts actually struggle and models can help) nor it is clear if it will actually be useful for the research community (models performing well on this dataset will need to focus on specific abilities which have not been studied in the research community).

---

> ### Author Response · Authors · 2018-01-05
> **Rebuttal (issues raised by Reviewer 3)**
>
> Thank you very much for your review and the constructive criticism.
> Please find below our responses (each below a brief summary of the corresponding issue):
>
> 1) Synthetic dataset, not clear if visual reasoning actually required or biases are exploited. A good test would be to evaluate the model on another task, e.g. on real figures.
> --> We recognize this and plan to address this issue by annotating images from the FigureSeer dataset (Siegel et al., 2016) and other sources in the future.
> We tried to address the bias concern in the original manuscript by balancing the data set. The text-only model does not perform better than chance. Since each image has exactly the same number of yes and no answers, a vision-only model can also not reliably achieve a higher performance than chance.
> The significant amount of annotation work required to produce such a test set with real images has prevented us from completing it for the original submission or our updated paper. Regardless of the source of these real figure images, the colors of the plot elements must be extracted or adjusted, which requires manual effort. We would need to crowdsource questions and answers for these images as well. Both of these aspects have been beyond our time and monetary budgets.
>
> 2) The only advantages of synthetic dataset mentioned in paper are greater control over task complexity and availability of auxiliary supervision signals, but this is not explored in the manuscript.
> --> We take inspiration from other visual reasoning datasets like CLEVR (Johnson et al., 2017) and NLVR (Suhr et al., 2017) for our task. Ultimately the effort to collect and annotate real figure images was prohibitively costly. Crowdsourcing efforts would be needed to extract colors, plot element names, and data points as well as to generate questions and answers - all of which have no accuracy guarantees.
> According to your suggestion, we have revised the manuscript to clearly state that experiments exploring additional annotations are outside of the scope of this paper and that the annotations are provided to encourage researchers to define other tasks. It also emphasizes that having reliable ground-truth targets is maybe the most important benefit of using a synthetic dataset and that weaknesses can be addressed by iteratively updating the corpus.
>
> 3) Need to discuss new challenges posed by FigureQA and high-level description of how to approach a solution to them.
> --> To perform well on the FigureQA task, models need to detect and reason about spatial properties and relationships between them. By our task formulation we want to encourage approaches for intuitive understanding of figures, that do not invert the visualization pipeline, i.e. do not revert to reconstructing the coordinates of data points. We think that end-to-end training of models on raw images and text on this task instead of training multiple disjoint components is more likely to adapt well to future extensions of this data set.
> Since our data includes questions aiming both at small visual details (e.g. dot-line plots) as well as larger patterns (e.g. area under the curve or pie slices), partially scale-invariant approaches may be useful. Standard CNNs are good at detecting patterns, but not great at detecting relations between them. Specialized architectures such as the Relation Network (Santoro et al., 2017) or FiLM (Perez et al., 2017) are more suited to visual reasoning tasks.
>
> 4) Why these 15 questions, are they the most useful questions for analysts? Why expect models to be good if humans already struggle with smoothness or median questions?
> --> Questions about maximum, minimum, median, area under the curve and the chosen plot types are often found in maths questions of GRE exams (example: https://www.ets.org/s/gre/accessible/gre_practice_test_3_quant_18_point.pdf, question 17). The goal in AI is to achieve human-level understanding and intelligence. Humans in GRE tests usually don’t create a table containing coordinates of data points before answering such questions, which is part of why we decided against including questions that require the reconstruction of quantities, such as coordinates of data points. As the research community gets closer to human performance in the FigureQA task, we plan to extend the data set with more question templates or crowd-sourced questions.
> The revised manuscript contains more detail on our motivation.
> [we had to split the rebuttal into two parts due to the character limit, this is the first half]

---

> > ### Author Response · Authors · 2018-01-05
> > **Rebuttal (issues raised by Reviewer 3, continued)**
> >
> > 5) Why only binary questions? It is probably unnatural for analysts to frame their problems in binary questions.
> > --> The choice of binary questions allowed us to balance the dataset to avoid problems with language biases as described in Goyal et al. (2016). NLVR (Suhr et al., 2017), another visual reasoning dataset, also poses a binary classification (is the provided statement true or false). We assume that a representation learned on a balanced binary dataset would at least be useful as a strong initialization for the non-binary setting and plan to investigate this in future work.
> >
> > 6) Why these 5 types of plots. Are they the most frequently used ones?
> > --> As mentioned in our response to 4), the dataset is in part inspired by math questions, such as those found in GRE exams. Besides scatter plots, these are the standard plot types in Matplotlib, the plotting library we used in the initial phase of development.
> > As the research community gets closer to a solution of the FiguraQA task, we plan to extend the dataset. Examples of interesting plot types would be scatter plots, Venn diagrams, area charts and radar plots, or compound types, such as line/area-bar charts or pareto charts.
> >
> > 7) Are accuracies of the models in Table 3 on the same subset that humans were tested on? If not, they should be reported separately.
> > --> Thank you for pointing this out. We agree that this should have been considered and revised the manuscript. It now separately reports the performance on the full test set in one table and compares CNN vs RN vs the human performance on a test subset in another table.
> >
> > References:
> > - Goyal, Yash, et al. "Making the V in VQA matter: Elevating the role of image understanding in Visual Question Answering." arXiv preprint arXiv:1612.00837 (2016).
> > - Johnson, J., Hariharan, B., van der Maaten, L., Fei-Fei, L., Zitnick, C. L., & Girshick, R. (2017, July). CLEVR: A diagnostic dataset for compositional language and elementary visual reasoning. In 2017 IEEE Conference on Computer Vision and Pattern Recognition (CVPR) (pp. 1988-1997). IEEE.
> > - Perez, Ethan, et al. "FiLM: Visual Reasoning with a General Conditioning Layer." arXiv preprint arXiv:1709.07871 (2017).
> > - Santoro, Adam, et al. "A simple neural network module for relational reasoning." arXiv preprint arXiv:1706.01427 (2017).
> > - Siegel, N., Horvitz, Z., Levin, R., Divvala, S., & Farhadi, A. (2016, October). FigureSeer: Parsing result-figures in research papers. In European Conference on Computer Vision (pp. 664-680). Springer International Publishing.
> > - Suhr, A., Lewis, M., Yeh, J., & Artzi, Y. (2017). A corpus of natural language for visual reasoning. In 55th Annual Meeting of the Association for Computational Linguistics, ACL.

---

> > > ### Comment · AnonReviewer3 · 2018-01-16
> > > **Final recommendation**
> > >
> > > After reading the rebuttal, I am still concerned about the contributions of the paper. The proposed dataset is automatically generated with limited complexity and no necessity of language understanding in its current form, as pointed by AR2. It is not reasonable to review the work based on what its state might be in the future, when language understanding might play a role. Therefore, it is important to show real-life applications of the dataset in the paper. I also fail to clearly see any new challenges being posed by this dataset which are currently not being studied; models such as RN and FiLM were already introduced before this dataset. I recognize that this paper is a first step towards encouraging research in pattern recognition in visual representation of data such as figures, hence I will keep my original rating.

---

### Author Response · Authors · 2018-01-05
**Rebuttal submitted (please check below each review)**

Thank your to all reviewers for your constructive criticism.
We have revised our manuscript and addressed the issues raised by each reviewer below the respective reviews.

A few general remarks:
- We added updated multi-GPU implementation of the RN and CNN+LSTM baselines and added the improved performances to the revised manuscript.
- We added a baseline using VGG features pretrained on ImageNet, described in the revised manuscript.
- We also now provide the validation accuracies in the performance table.
- As mentioned in the new version, we will make the source code for all models available.

---

### Decision · Program_Chairs · 2018-01-29
**ICLR 2018 Conference Acceptance Decision**

**Decision:**

Invite to Workshop Track

**Comment:**

This paper was reviewed by 3 expert reviews. While they all see value in the new task and dataset, they raise concerns (templated language, unclear what exactly are the new challenges posed by this task and dataset, etc) that this AC agrees with. To be clear, the lack of a fundamentally new model is not a problem (or a requirement for every paper introducing a new task/dataset), but make a clear compelling case for why people should work on the task is a reasonable bar. We encourage the authors to incorporate reviewer feedback and invite to the workshop track.